# miniCodeProps: a Minimal Benchmark for Proving Code Properties

**Evan Lohn**
Carnegie Mellon University
`evanlohn@cmu.edu`

**Sean Welleck**
Carnegie Mellon University
`wellecks@cmu.edu`

## Abstract

AI agents have shown initial promise in automating mathematical theorem proving in proof assistants such as Lean. The same proof assistants can be used to verify the correctness of code by pairing code with specifications and proofs that the specifications hold. Automating the writing of code, specifications, and proofs could lower the cost of verification, or, ambitiously, enable an AI agent to output safe, provably correct code. However, it remains unclear whether current neural theorem provers can automatically verify even relatively simple programs. We present `miniCodeProps`, a benchmark of 201 program specifications in the Lean proof assistant, aimed at the subproblem of automatically generating a proof for a provided program and specification. `miniCodeProps` contains specifications about simple, self-contained programs (e.g., lists, natural numbers, binary trees) with varied proof difficulty. Despite its simplicity, `miniCodeProps` is sufficient to break current LLM-based provers, with state-of-the-art methods showing promise on the easy properties in `miniCodeProps`, yet failing to prove nearly all of the medium and hard properties. We publicly release `miniCodeProps` as a benchmark for furthering automated theorem proving in the context of formally verified code.[1]

## 1   Introduction

Writing code that meets a specification is a desirable, yet difficult task. In safety-critical contexts, users require stronger safeguards than code reviews and test cases provide. To address this need, the formal methods community has produced a variety of tools for specifying and proving properties of code [1–5]. One approach uses interactive theorem provers (ITPs) such as Isabelle [6], Coq [7], and Lean [8]. In an ITP, a user writes code, a specification, and a proof that the specification holds (Figure 1). The ITP checks each step of the proof as it is written, and a complete proof means that the specification is guaranteed to hold. A complete proof thus gives guarantees on the underlying code, which is valuable in practice. For instance, the Lean proof assistant was recently used to verify properties of Amazon's Cedar Policy language [9]. Ambitiously, future code generation models could also output safe, provably correct code if they are able to write code, specifications, and proofs [10].

Despite its promise, interactive theorem proving remains a difficult task, and better automation–be it writing specifications, proofs, or combinations thereof–could help make it easier to verify code. Recently, machine learning techniques based on large language models (LLMs) have shown promise in automatically generating code [11], proving mathematical theorems in ITPs [12], and generating proofs of properties from verification projects [13]. However, evaluating the capabilities of LLMs for program verification remains a challenge. For example, benchmarks often evaluate across projects with complex dependencies such as the CompCert compiler or Archive of Formal Proofs [14, 13], making it difficult to isolate and reason about a model's core capabilities and weaknesses. Rather than

---

[1]Our benchmark and evaluation code can be found here: https://github.com/cmu-l3/minicodeprops-eval

38th Conference on Neural Information Processing Systems (NeurIPS 2024) Safe Generative AI Workshop.

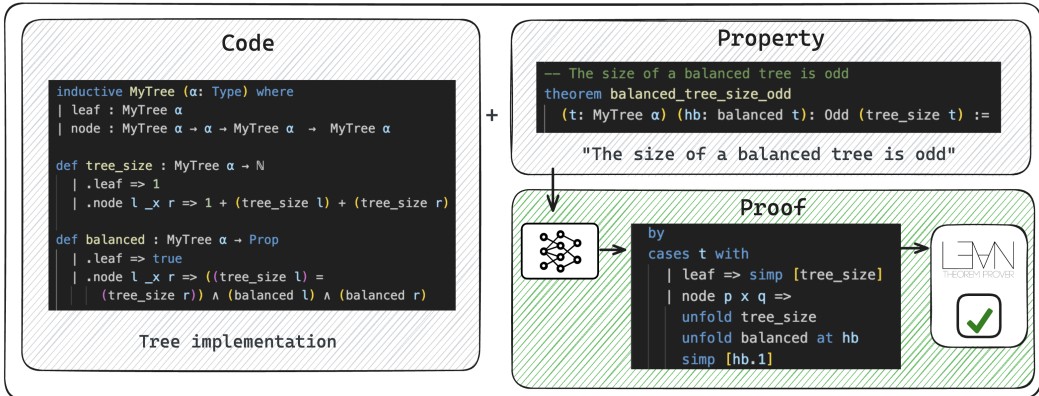

Figure 1: Formal verification of code in an interactive proof assistant (here, Lean) consists of writing (i) code (e.g., an implementation of a Tree), (ii) a property that you want to verify (e.g. that the size of a balanced tree is odd), and (iii) a proof that the code satisfies the property. The underlying language verifies that the proof is correct, providing certainty that the property holds. `miniCodeProps` focuses on the important subproblem of proof generation: given a property and associated code, a model must generate a proof (e.g., shown here in green) that the property holds. `miniCodeProps` contains 201 properties about lightweight, self-contained code blocks, and measures a range of proving abilities.

evaluating on complex projects, our goal is to find a minimal set of meaningful program properties that breaks current LLM-based provers, and thus requires new methods or models to solve.

To this end, we introduce `miniCodeProps`, a benchmark for evaluating the ability to prove properties of relatively simple, self-contained programs in the Lean proof assistant. `miniCodeProps` contains 201 specifications about simple, self-contained programs (involving e.g., lists, natural numbers, binary trees) with varied proof difficulty, sourced by translating Haskell programs from Tons of Inductive Programs [15] into Lean 4. Despite its simplicity, `miniCodeProps` is challenging for current LLM-based provers. For example, our best baseline approach based on GPT-4 proved very few specifications requiring proofs longer than a few lines. We publicly release `miniCodeProps` as a benchmark for furthering automated theorem proving in the context of formally verified code.

## 2   Related Work

**Automating mathematical theorem proving.**   Recently there has been wide interest in automating mathematical theorem proving in interactive proof assistants; see [16, 12] for surveys. A typical approach [17] is to train on a large corpus of mathematical proofs such as Lean's Mathlib [18–22]. A model learns to call automation that is designed for mathematics, such as nonlinear arithmetic tactics, and to use definitions and theorems from within the corpus. It is unclear whether such methods transfer to the distribution of proofs encountered in program verification, which may rely on calling different automation, using program-specific definitions or lemmas, or different proof strategies. By focusing on Lean, `miniCodeProps` allows for testing this transferability. A second class of methods design prompting strategies with mathematical problems in mind, such as conditioning generation on an informal proof [23–25] or examples from Mathlib [26]. We hope that `miniCodeProps` motivates development of similar methods for program verification.

**Automating formal code verification.**   There is a rich history of developing automation for formal verification of code; see [4] for a survey. For machine learning in interactive theorem proving, many methods that preceded large language models focused on Coq, such as ProverBot [27, 28], ASTactic [14], TacTok [29], Diva [30], and Passport [31]. Recent work with LLMs includes exploring prompting strategies in Coq [32], and Baldur [13] which explores proof generation and repair in Isabelle's Archive of Formal Proofs. We are not aware of machine learning-based proof automation targeting program verification in Lean, despite the activity in automated mathematical proving in

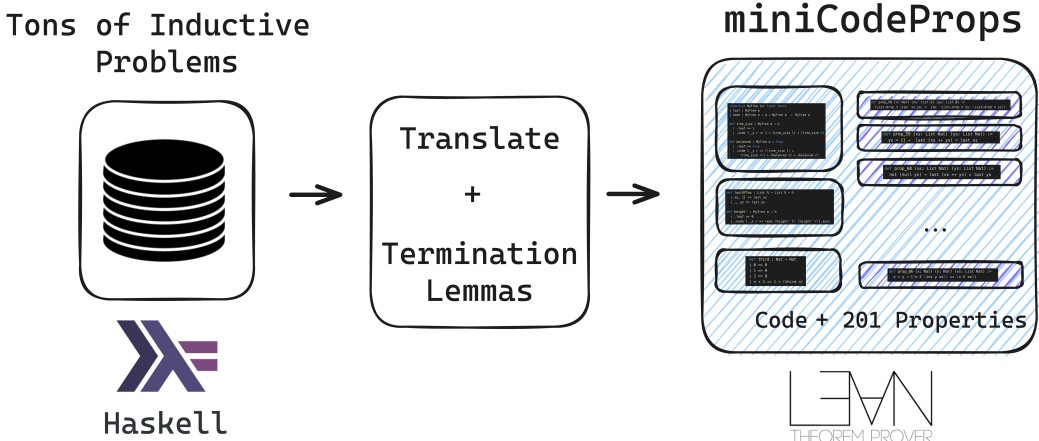

Figure 2: `miniCodeProps` is sourced from Tons of Inductive Problems [15], a collection of programs and specifications written in Haskell. We translate the programs into Lean, and write and prove termination lemmas that are needed to state and prove properties of recursive functions.

Lean. One of our motivations is bridging this gap. Finally, some methods such as COPRA [25] aim to be language-agnostic; `miniCodeProps` could participate in evaluating and developing such systems.

A second paradigm of verification is based on automated reasoning (SAT/SMT-based) languages such as Dafny [1] and Verus [5] (Rust). Recent work explores LLM-based automation in these languages, such as Clover [10] for Dafny, and [33] for Verus. These are complementary to studying automated ITP verification. Finally, Tons of Inductive Problems [15]–from which we derive our data–targets automated reasoning-based verification, while `miniCodeProps` targets interactive theorem proving.

**Interactive theorem proving benchmarks.** The CoqGym [14] benchmark tests on over 100 repositories in Coq, while some papers use code properties from a single large, complex repository such as the CompCert compiler verification project in Coq [34, 25], which arguably tests different aspects of automated code verification than those tested by `miniCodeProps`. In Isabelle, the Archive of Formal Proofs contains some verification-related sections that are used for evaluation, e.g., in Baldur [13]. Benchmarks for automated theorem proving in Lean are focused on theorems from mathematical domains. For example, miniF2F [35] contains 488 self-contained, easy-to-state theorems from math competitions, while ProofNet [36] contains self-contained textbook problems. miniF2F's simplicity and impact as a benchmark motivated the creation (and naming) of `miniCodeProps`.

## 3   Benchmark Contents

We create `miniCodeProps` by translating programs from Tons of Inductive Programs (TIP) [15] (https://github.com/tip-org/benchmarks) from Haskell into Lean 4. We manually translated code from three files in TIP that contain a mix of function definitions, propositions describing the results of calling those functions, and termination lemmas used in the function definitions. In total, this yields 201 Lean 4 theorem statements. We specifically make `miniCodeProps` fairly small, since evaluating theorem proving models requires many calls to the language model (e.g., in a tree search or through repeated sampling), and is hence time-consuming and prohibitively expensive for large datasets.

### 3.1   Collection Process

The properties in TIP describe induction in both programming and purely mathematical contexts. Since our focus is programming, we filtered out files that contained primarily mathematical induction, then translated the remaining Haskell code into Lean 4. We associated each property in the translated files with dependencies and metadata useful for different evaluation modes of the benchmark.

| | Numbers | Lists | Trees | Heaps | Difficulty |
|---|---|---|---|---|---|
| Medley Functions | 0 | 26 | 10 | 0 | – |
| Medley Properties | 21 | 84 | 5 | 0 | Easy |
| Termination+Sorting Functions | 3 | 49 | 2 | 5 | – |
| Termination Properties | 3 | 22 | 0 | 3 | Medium |
| Sorting Properties | 0 | 63 | 0 | 0 | Hard |

Table 1: **miniCodeProps Composition** We classified each function and property in the Lean source files by their primary subject matter. We also separate these numbers by the splits of `miniCodeProps`: Medley, Termination lemmas, and Sorting algorithms. Termination and Sorting algorithm functions are grouped because the properties in both splits describe the same set of functions. The properties are classified into easy (Medley), medium (Termination), and hard (Sorting) difficulty levels.

## 3.2 Source Code Composition

`miniCodeProps` contains properties that we view as a "minimum level of competency" for automated neural theorem provers. We draw an analogy to an expert software engineer that spends time writing code in a complex repository, yet is expected to be capable of solving self-contained interview problems. `miniCodeProps` embodies the latter, which is absent from existing theorem proving benchmarks. Namely, rather than pulling code and properties from complex verification projects, `miniCodeProps` tests key capabilities with lightweight, self-contained code blocks and properties.

These fall into three categories: (1) intuitive properties of lists, trees, and natural numbers (*medley*), (2) lemmas that support termination proofs, which are an essential (and tedious) part of verification in an interactive theorem prover (*termination*); (3) properties of nonstandard sorting algorithms that require a deep understanding of the corresponding code (*sorting*). We provide more detail below.

**Functions.** Most of the translated functions operate on linked lists, while the rest involve natural numbers and binary trees. With a few notable exceptions, the functions perform conceptually simple operations such as filtering, returning the last element, and counting the elements of a list. The more complicated functions are increasingly esoteric sorting functions. In total, `miniCodeProps` is derived from 95 TIP functions. A breakdown is provided in Table 1.

**Properties.** The properties to be proven in `miniCodeProps` express intuitive properties of the function(s) being described. Properties in the *medley* classification typically require a standard induction argument along with the application of 1-2 library theorems. Properties of the nonstandard sorting algorithms are easy to state, yet are likely much more difficult prove due to the complexity of the sorting functions. See Section 3.3 for examples, and Table 1 for a breakdown by subject type.

**Termination Lemmas.** In Lean 4, recursive functions must be paired with a proof of termination. While Lean automatically infers the proof in simple cases, more complicated recursive calls require the user to explicitly prove termination. As the vast majority of our function definitions are recursive, our benchmark includes 28 lemmas that support 4 termination proofs of nonstandard sorting algorithms from TIP. These lemmas represent highly nontrivial properties of code that, roughly speaking, may take hours of human effort to prove. Automating these proofs would be highly desirable in practice.

**Dataset format.** The benchmark is formatted as a jsonlines file with the information in Table 2 per line. The "full_name" property is used to determine the split the property belongs to: *medley* properties are named "prop_n" for $n \in [1, 86]$, *sorting* algorithm properties are those remaining theorems named beginning with "prop", and the remainder are *termination* lemma properties.

## 3.3 Qualitative Examples

We study several benchmark examples and describe what makes them interesting and difficult.

**Zip over concatenation.** We begin with a reimplementation of the standard 'zip' function that combines two lists into a single list of paired elements. 'prop_84' states that zipping a list $xs$ with the concatenation of two other lists $ys, zs$ is equivalent to separately zipping appropriate sections of

Table 2: Example code property and associated data

| Key | Description | Examples |
|---|---|---|
| full_name | Unique name of the property in the benchmark | 'prop_60' |
| prop_defn | The text of the code property to be proven | ```lean
lemma prop_60 (xs: List Nat)
(ys: List Nat) : not (null ys)
→ last (xs ++ ys) = last ys:=
by sorry``` |
| prop_loc | The file name and line number where the property definition begins | LeanSrc/LeanSrc/Properties.lean:257 |
| score | An integer in the range [1-5] used as a difficulty heuristic | 5 |
| deps | Text containing all the dependencies (functions and lemmas) that the property requires to be fully defined | ```lean
import Mathlib
def last: List Nat → Nat
  | [] => 0
  | [x] => x
  | _x::xs => (last xs)
def null : List α → Bool
  | [] => True
  | _ => False``` |
| proof_state | The initial proof state of the property | ```lean
xs ys : List ℕ
⊢ (!null ys) = true →
last (xs ++ ys) = last ys``` |
| file_locs | A list of the last line in each file needed to define the property. Appending each file up to this line before the property fully defines the property. | [["LeanSrc/LeanSrc/Definitions.lean", 249], ["LeanSrc/LeanSrc/Properties.lean", 258]] |

$xs$ with each of $ys$ and $zs$, then concatenating the results. To the best of our knowledge, Mathlib does not contain an equivalent lemma for the standard definition of the zip function.

```lean
def zip' : List α → List β → List (α × β)
  | [], _ => []
  | _, [] => []
  | x::xs, y::ys => ⟨x, y⟩ :: zip' xs ys

lemma prop_84 (xs: List α) (ys: List β) (zs: List β) :
  (zip' xs (ys ++ zs) =
  zip' (List.take (List.length ys) xs) ys ++
  zip' (List.drop (List.length ys) xs) zs):= by
```

This property is intuitively correct, but requires some intuition about proofs of recursive structures to correctly prove in Lean. The following is a correct proof of prop_84 generated by GPT4. There are several unnecessary definitions passed to the calls to the simplifier (the 'simp' tactic). However, GPT4 demonstrates the main ideas of the proof: induct on $ys$, pair off the first elements of $xs$ and $ys$, then apply the inductive hypothesis to what remains.

```lean
induction ys generalizing xs with
  | nil => simp [zip', List.take, List.drop]
  | cons y ys ih =>
```

```
    cases xs with
    | nil => simp [zip', List.take, List.drop]
    | cons x xs =>
      simp [List.cons_append, zip', List.take,
        List.drop, List.length]
      rw [ih]
```

Many of the *medley* properties test similar abilities, which we view as bare minimum capabilities for a system aimed at automating theorem proving in the context of formally verified code.

**Non-standard insertion increments length.**    Although the zip concatenation property is not part of Mathlib, one can argue that because the property involves a utility function defined in Mathlib with its own associated properties and proofs, it is not altogether surprising that GPT4 was able to infer a proof of a novel property of a re-definition of the function. We therefore present a property of a function 'ins' not included in Mathlib that inserts an element into a list before the first element it is less than. Proving this property requires the ability to reason about previously unseen code.

```
def ins: Nat → List Nat → List Nat
  | n, []   =>  [n]
  | n, x::xs => if (n < x) then n :: x :: xs else x :: (ins n xs)
```

The following is a correct proof generated by GPT-4. This proof is remarkable for its simplicity. Unlike the previous correct proof there is little wasted space, a feature often sought after by human proof experts. GPT4 demonstrates understanding of how to work with the 'ins' function via the 'split_ifs' tactic, and the usage of the inductive hypothesis only in the second branch where it is necessary.

```
induction xs with
| nil => simp [ins]
| cons y ys ih =>
  simp [ins]
  split_ifs with h
  { simp }
  { simp [ih] }
```

**Example sorting properties.**    The Lean source code of `miniCodeProps` contains implementations of 9 sorting algorithms, with associated properties to prove for each algorithm. We display several of the functions and lemmas associated with one of our two heapsort implementations (each uses a different strategy for merging heaps). The 'toList' function contains an example of a termination lemma: the line beginning with 'have _h' references a lemma that is used by Lean to prove that the recursion in 'toList' terminates, a property required by default for recursive functions in Lean. Several of the functions and lemmas referenced are omitted from the code for brevity.

```
def toHeap : List Nat → MyHeap
| xs => hmerging (xs.map (fun x => MyHeap.node MyHeap.nil x MyHeap.nil))

lemma numElem_merge_branches_lt (p q: MyHeap) (x: Nat): numElem (hmerge p q) <
    numElem (MyHeap.node p x q) := by
  rw [←merge_elems _ _];
  have h': numElem (MyHeap.node p x q) = 1 + numElem p + numElem q; rfl
  rw [h']
  linarith;

def toList : MyHeap → List Nat
| MyHeap.nil => []
| MyHeap.node p x q =>
    have _h := numElem_merge_branches_lt p q x
    x :: toList (hmerge p q)
termination_by hp => numElem hp

def hsort : List Nat → List Nat
  | xs => toList (toHeap xs)
```

```
theorem prop_HSortSorts (xs: List Nat) : ordered (hsort xs) == True := by sorry
theorem prop_HSortCount (x: Nat) (xs: List Nat) : count x (hsort xs) == count x xs
    := by sorry
theorem prop_HSortPermutes (xs: List Nat) : isPermutation (hsort xs) xs == True :=
    by sorry
theorem prop_HSortIsSort (xs: List Nat) : hsort xs == isort xs := by sorry
```

The properties above express the following intuitive properties: heapsort orders list elements, preserves the list size, returns a permutation of the original list, and returns the same result as insertion sort. These properties are repeated for each sorting algorithm. However, given the complexity of the sorting functions involved, we expect that proving such properties will be very difficult. An example of an incorrect proof of one of the sorting properties produced by GPT4 can be found in Appendix A.

Finally, `miniCodeProps` includes termination lemmas such as 'numElem_merge_branches_lt' with the human-produced proof removed. Some of these lemmas took hours of human effort to prove. Since proving termination is an essential part of human interactive theorem proving, it is highly likely that improved performance on these lemmas could translate into useful automation for practitioners.

**Availability and usage.** We have published our benchmark as a public dataset on Huggingface, and released a repository for evaluating language models on `miniCodeProps`.

## 4 Baselines

We test two standard proving modes: full-proof generation and tactic-by-tactic generation. Since the underlying code block for a property is typically necessary to complete the proofs, we always provide the models with the code dependencies (stored as "deps" in our dataset, see Table 2).

For full-proof generation, the model generates one or more potential proofs that are checked by the proof checker (in our case, the Lean 4 kernel). For tactic-by-tactic generation, we follow the common practice of taking the proof state as input and returning suggestions for the next tactic to use in the proof. Each suggestion is then given as input to the Lean 4 kernel, and the resulting proof state is used to prompt the model for the next tactic. We test the *de-facto* standard algorithm for tactic generation, best-first search [17, 19, 22, 26]. Our publicly available evaluation code supports both modes.

**Tactic-by-tactic generation.** We use `ntp-ctx-1.3`B [37] since at the time of experimentation it is the only tactic generation model specifically trained to accept both the current proof state and an additional input context (in our case, the necessary underlying code for the property being proven). The `ntp-ctx-1.3`B model is a DeepSeek-Coder 1.3B language model fine-tuned on (context, proof state, next tactic) examples derived from Mathlib [37]. For the context, following a similar setup to [37] we provide a context containing the dependent code ("deps" in Table 2), the property being proven, and the history of generated tactics. We use the same input format and search settings as [37].

**Full proof generation.** We use GPT-4o, since at the time of experimentation it is a highly-performant general-purpose language model on mathematical and code tasks, and its model variants (e.g., previous models of GPT-4) are common baselines in neural theorem proving (e.g., [25, 37]). We provide a context containing the dependent code ("deps" in Table 2) and the property being proven. We measure pass@32, meaning that we generate 32 proof candidates (with default generation parameters), and consider a property to be successful when at least one of them passes the verifier.

**Full proof generation with refinement.** As a strictly stronger full proof generation baseline, we perform 2 rounds of an experimental "many-to-one" refinement technique using GPT-4o. Specifically, suppose that none of the 32 candidate proofs $y^{(1)}, \ldots, y^{(32)}$ passed the verifier. We provide all 32 candidate proofs in a prompt, and ask the model to learn from the failed attempts and generate a new proof. We sample $k$ such generations, which constitutes one round of refinement. We do this for $T$ rounds, feeding in only the previous round's generations. We use $k = 16$ and $T = 2$. In a small scale experiment we confirmed that this outperformed sampling 64 candidates without refinement.

## 5 Results and Discussion

**Results.** Our baseline results are displayed in Table 3. All three baselines show nontrivial performance on the easy (*medley*) properties. This shows that current neural theorem provers have some

| Model | Medley (Easy) | Termination + Sorting (Medium + Hard) | Overall (All) |
|---|---|---|---|
| GPT-4o (pass@32) | 75.6% (65/86) | 4.34% (5/115) | 34.8% (70/201) |
| + refinement | 77.9% (67/86) | 6.96% (8/115) | 37.3% (75/201) |
| ntp-ctx-1.3B [37] | 72.1% (62/86) | 8.69% (10/115) | 35.8% (72/201) |

Table 3: **miniCodeProps results.** Number of specifications proven using full proof generation with GPT4 and best-first tactic generation with `ntp-ctx-1.3B` to the problem of verifying program specifications. The Medley section contains mostly of specifications that can be proven in several lines. Proofs of the sorting algorithm properties and termination lemmas are expected to require at least tens of lines and hours of programmer effort.

ability to perform basic inductive proofs in a program verification setting. However, the three methods perform very poorly on the medium and hard properties. The results show that `miniCodeProps` is sufficient to break current neural theorem proving methods, hence motivating the need for further research. Moreover, it is unlikely that these high-level conclusions would change given more properties drawn from the same distribution, thereby justifying `miniCodeProps`' size of 201 properties.

However, it is promising that changing the generation algorithm (namely, to refinement) leads to some additional successes on the medium and hard split. This suggests that further algorithmic development could lead to additional gains. Furthermore, the small `ntp-ctx-1.3B` model trained on Mathlib shows comparable, or better, performance than GPT-4o. As a result, we view training specialized models for verification-style proofs as a promising direction as well (though requiring nontrivial research due to the scarcity of training data). Based on these observations, we speculate that neural theorem provers have the potential to improve on `miniCodeProps` in the near future.

**Scope.** We chose to source `miniCodeProps` from the TIP dataset to provide some assurance that the code and properties involved were of interest to the broader code verification community. Clearly, performing well on this subset is only a proxy for the ability to perform well on all realistic code properties. However, we designed `miniCodeProps` to target a minimal set of code properties that we would expect a reasonable automated system to be able to prove. This is similar to an expert software engineer that spends time working in large, complex codebases, yet is expected to be capable of solving self-contained exercises that target core programming skills.

**Societal context.** The broader goal of our work is to facilitate the development of automated (likely, neural) theorem proving agents that can prove properties of code in ITPs. Verification engineers already draw from a wide variety of automated tools in their work. An automated version of ITP still requires a human in the loop to generate (or validate) the statement of the property to be proven, similar to the type of work these engineers perform while using SMT-based verification. Arguably, the main benefit to society of the ITP approach is that it induces our automated tools to use different reasoning patterns than the ones commonly used in Automated Reasoning (SAT/SMT) approaches, which may in turn lead to different subsets of formal guarantees that can be obtained in practice.

## 6 Conclusion

`miniCodeProps` is intended as a meaningful benchmark for evaluating techniques that automate ITP-based code verification, including for safe generation of code by AI agents. It contains a diverse array of functions and properties from Tons of Inductive Problems, allowing the possibility of incremental progress in verifying the types of programs contained in TIP. We show that simple baseline approaches from mathematical neural theorem proving currently fall short on much of our benchmark, and hope that `miniCodeProps` spurs development of future ITP code verification agents.

**Acknowledgements**

Sean Welleck thanks Convergent Research, the Lean FRO, and the OpenAI Researcher Access Program for their support.

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

# A  Appendix

This is an incorrect proof of the 'heap sort orders list elements' property. GPT4 leaves in "placeholders" as arguments to lemmas from the environment that it does not use in the proof. It also attempts the invalid operation of induction generalizing the inductive type 'MyHeap'. Transforming this proof into a correct one is non-trivial.

```
unfold hsort
unfold ordered
induction xs generalizing MyHeap with
| nil => unfold toHeap; unfold toList; simp
| cons x xs' ih =>
  unfold toHeap
  unfold toList
  unfold hmerge
  have h := numElem_merge_branches_lt ... -- Placeholder for specific heaps
  have m := merge_elems ... -- Placeholder for specific heaps
  specialize ih (xs'.map (fun x => MyHeap.node MyHeap.nil x MyHeap.nil))
  induction (toList (toHeap xs')) generalizing (toList (toHeap (x :: xs')))
  | nil =>
    apply ih
  | cons y ys' ysih =>
    unfold toList at ih
    unfold toHeap at ih
    unfold ordered at ih ⊢
    unfold hsort at ih
    have h := numElem_merge_branches_lt ... -- Placeholder for specific values
    have m := merge_elems ... -- Placeholder for specific values
    case h_1 =>
      unfold ordered
      simp [List.map]
      simp [toList]
    case h_2 =>
      unfold toHeap
      unfold ordered
      simp [toList]
      rfl
```

