# OpenReview forum: "miniCodeProps: a Minimal Benchmark for Proving Code Properties"
_NeurIPS.cc/2024/Workshop/SafeGenAi — SafeGenAi Poster_

### Official Review · Reviewer_iwBr · 2024-10-08
**A good benchmark for LLM assessment for formal proofs**

**Rating:** 7
**Confidence:** 4

**Review:**

Summary
========
The authors propose miniCodeProps: A small benchmark of 201 program specifications, compatible with the Lean proof-assistant, that is meant to assess LLM capabilities in generating formal proofs for verifying whether some code adheres to certain specifications. To do so, the authors manually translate programs for Tons of Inductive Programs (TIP) written in Haskell to Lean 4. Rather than pulling complex properties, the authors design lightweight, self-contained properties for the dataset. These categories test properties of lists etc (medley), termination proofs, and sorting-based properties. The authors also include a discussion that include properties not existing in any standard benchmark and thus not susceptible to memorization at the time of writing.

The authors then showcase the value of their benchmark by conducting an evaluation in two modes: (a) full-proof generation: Generate the full proof all at once, (b) tactic-by-tactic generation. The authors use gpt-4o for (a) and use a compatible ntp-ctx model for (b). They measure pass@32 scores with any success counted as a success. The authors show that current SOTA LLM technology is not sufficient for non-trivial tasks.

The authors also show an iterative approach of iterative refinement that seems to improve results on hard problems.

Review
=======
This paper presents a useful benchmark and targets an important problem for formal proof verification and program synthesis that is of great interest to the safegenai community. The paper is well-written, the ideas are presented well and the benchmark suggested clearly indicates the lack of SOTA LLMs in safely and robustly generating valid proofs.

This paper would of great use to spur discussion on the LLMs and their use in proof-assistants and co-pilots spanning the formal proof (SMT)-etc and program verification communities.

---

### Official Review · Reviewer_ocLJ · 2024-10-08

**Rating:** 6
**Confidence:** 3

**Review:**

**Short Summary:** This paper introduces a new benchmark miniCodeProps for evaluating LLMs and different approaches for generating proofs verifying self-contained code for a specified property.

**Quality and Clarity:** Overall, the points made by the paper were clear and well-written.

**Originality:** To my knowledge, this paper provides a novel benchmark for measuring the ability to generate proofs for a given program and property to prove. There have been benchmarks created using theorem provers for evaluating LLMs for other tasks (autoformalization, etc.), but none have been created for evaluating an LLM’s capability for generating proofs on verifying code specifications.

**Strengths:**
- The paper tackles an interesting problem and justifies why this task could be used with automatic proof synthesis to verify code.
- The paper provides extensive details on the dataset they constructed for miniCodeProps, with an analysis that breaks down the composition of the problems.

**Weaknesses:**
- Details on the exact approach taken to performing the miniCodeProps benchmark are not included beyond a couple of paragraphs discussing how the benchmark was calculated for each individual baseline.
- Little analysis is done on the benchmark results, making it unclear what each baseline approach's strengths and weaknesses are beyond its score on each problem category.
- I am concerned about whether comparing the accuracy of the three baselines discussed in this paper is a fair comparison. The score for full proof generation represents the number of examples where one of the 32 proof candidates passes compared to tactic-by-tactic, where the score represents whether tactic steps were found to generate the proof. For refinement, this score represents the ability to generate the correct proof given the additional knowledge of failed attempts if the 32 proof candidates are inaccurate.

**General Review:** This paper is well-written and discusses an interesting problem that can be applied to verifying the code generation of other LLMs. It also evaluates LLMs on their ability to generate proofs in this context. While I have concerns about the evaluation (see Weaknesses above), I will give it an acceptance.